# Serological Investigation and Genetic Characteristics of Pseudorabies Virus between 2019 and 2021 in Henan Province of China

**DOI:** 10.3390/v14081685

**Published:** 2022-07-30

**Authors:** Ximeng Chen, Hongxuan Li, Qianlei Zhu, Hongying Chen, Zhenya Wang, Lanlan Zheng, Fang Liu, Zhanyong Wei

**Affiliations:** 1International Joint Research Center of National Animal Immunology, College of Veterinary Medicine, Henan Agricultural University, Zhengdong New District Longzi Lake 15#, Zhengzhou 450046, China; simonarchimonde@126.com (X.C.); li17836997254@126.com (H.L.); zhll2000@sohu.com (L.Z.); liufang.vet@henau.edu.cn (F.L.); weizhanyong@henau.edu.cn (Z.W.); 2Henan Center for Animal Disease Control and Prevention, Zhengzhou 450002, China; zhuqianlei123@126.com; 3Key Laboratory of “Runliang” Antiviral Medicines Research and Development, Institute of Drug Discovery & Development, Zhengzhou University, Zhengzhou 450001, China

**Keywords:** Aujeszky’s disease, pseudorabies virus, epidemiological characteristics, phylogenetic analysis

## Abstract

In late 2011, severe pseudorabies (PR) outbreaks occurred among swine herds vaccinated with the Bartha-K61 vaccine in many provinces of China, causing enormous economic losses for the pork industry. To understand the epidemic profile and genetic characteristics of the pseudorabies virus (PRV), a total of 35,796 serum samples were collected from 1090 pig farms of different breeding scales between 2019 and 2021 in the Henan province where swine had been immunized with the Bartha-K61 vaccine, and PRV glycoprotein E (gE)-specific antibodies were detected using an enzyme-linked immunosorbent assay (ELISA). The results reveal that the overall positive rate for PRV gE antibodies was 20.33% (7276/35,796), which decreased from 25.00% (2596/10,385) in 2019 to 16.69% (2222/13,315) in 2021, demonstrating that PR still existed widely in pig herds in the Henan province but displayed a decreasing trend. Further analysis suggested that the PRV-seropositive rate may be associated with farm size, farm category, quarter, region and the cross-regional transportation of livestock. Moreover, the gE gene complete sequences of 18 PRV isolates were obtained, and they shared a high identity (97.1–100.0%) with reference strains at the nucleotide level. Interestingly, the phylogenetic analysis based on the gE complete sequences found that there were both classical strains and variant strains in pig herds. The deduced amino acid sequence analysis of the gE gene showed that there were unique amino acids in the classical strains, the variant strains and genotype Ⅱ strains. This study provides epidemiological data that could be useful in the prevention of pseudorabies in Henan, China, and this finding contributed to our understanding of the epidemiology and evolution of PRV.

## 1. Introduction

Porcine pseudorabies (PR), also called Aujeszky’s disease, is an acute infectious disease with high morbidity and mortality which has caused great harm to the global animal husbandry industry [1]. The causative agent, suid herpesvirus type 1 (SuHV-1, syn. Aujeszky’s disease virus or pseudorabies virus (PRV)), is an enveloped and double-stranded linear DNA virus which is taxonomically classified into the family *Herpesviridae*, subfamily *Alphaherpesvirinae*, genus *Varicellovirus* [2]. Members of the family Suidae (true pigs), as unique natural hosts and reservoirs of PRV, are susceptible to PRV at all ages [3], with clinical symptoms including lethal encephalitis and neurological symptoms in neonatal piglets, respiratory disease in finishing pigs and reproductive failure in infected sows [4,5]. Without specific host tropism, PRV was previously deemed to infect a wide range of mammals including ruminants, carnivores and rodents, with the exception of higher-order primates and humans [6,7,8]. Nonetheless, multiple reports have shown that PRV can cross the livestock-to-human species barrier, invade the human central nervous system (CNS) and induce human encephalitis [9,10,11,12], in the case of a PRV strain isolated from acute human encephalitis, suggesting that humans may be a potential host for PRV [13]. Besides, there are currently no effective drugs to prevent and treat the disease, which could pose a potential threat to public health.

In China, PR was first reported in the 1950s, and was well-controlled between 1990 and 2011 due to the widespread use of glycoprotein E (gE)-negative vaccines based on the Hungarian strain Bartha-K61 [14]. However, at the end of 2011, PR outbreaks took place on many Chinese farms where swine had been immunized with the Bartha-K61 vaccine, and rapidly spread to many regions of China, causing a significant impact on the pig-farming industry [15,16,17,18]. Studies indicated that the re-emerging PR was caused by PRV variants, and Bartha-K61 vaccines only confer partial protection on piglets against these new variants [15,17,19,20]. It is reported that about 2600 newborn piglets and 200 sows died from the PRV variant infections in one swine herd in the Henan province, resulting in a direct economic loss of at least one million Chinese yuan (CNY 156,000) [21].

To better prevent and control the disease, serological and molecular epidemiology investigations of PRV infection were carried out extensively in the Henan province of China before 2019 [18,19,22], but its epidemic profile in recent years remains unclear. Hence, the current study was designed to investigate the seroprevalence of PR from 2019 to 2021 in the Henan province of China and to analyze the gE gene of PRV strains isolated in this study.

## 2. Materials and Methods

### 2.1. Sample Collection

From January 2019 to December 2021, a total of 35,796 serum samples were collected from 35,796 pigs on 1090 farms in the Henan province of China (Figure 1). The 1090 pig farms consisted of three different categories: slaughterhouses, commercial pig farms and breeding farms. Owing to various breeding scales of stock farms, 15∼29, 30 and 31∼200 samples were collected from each small- (<500 pigs), medium- (500∼2000 pigs) and large-scale farms (>2000 pigs), respectively. Each batch of serum samples was collected on the day after blood collection and stored at −80 °C, and the specific detection was completed the next day. The results from different cities will be aggregated into the database monthly. Additionally, 389 tissue samples containing brains, lymph nodes and lungs were collected from 389 diseased pigs during outbreaks of PR. These tissue samples were homogenized, diluted at a ratio of 1:5 in phosphate-buffered saline (PBS) and then stored at −80 °C until use.

### 2.2. Serological Detection

Serological analysis was performed to detect anti-gE antibodies (Abs) using commercial blocking enzyme-linked immunosorbent assay (ELISA) kits (Pseudorabies Virus gE Antibody Test Kit) (IDEXX Laboratories, Westbrook, ME, USA), according to the manufacturer’s procedure. Results were calculated by dividing the absorbance at 650 nm (A (650)) of the tested sample by the mean A (650) of the negative control, resulting in a sample/negative (S/N) value that was used to differentiate infected from vaccinated animals. The S/N value was inversely proportional to the quantity of Abs. Therefore, S/N ≤ 0.60 was considered as positive, S/N > 0.70 was regarded as negative, and 0.60 < S/N ≤ 0.70 was judged to be suspicious, necessitating that the test be retested or retested over time.

### 2.3. Virus Detection and Isolation

The tissue samples were freeze–thawed three times to release the virus, and the homogenates were centrifuged at 8000× *g* for 5 min. Viral DNA was extracted from the tissue samples using the Nucleic Acid Extraction and Purification Kit (Omega Bio-Tek, Inc., Norcross, GA, USA) according to the manufacturer’s instructions. The presence of PRV nucleic acids was screened by polymerase chain reaction (PCR) as described previously [22]. PRV-positive tissue supernatants were filtered through a 0.22 μm filter (EMD Millipore, Billerica, MA, USA), and then inoculated into swine testis (ST) cell monolayers for 2 h in a 37 °C incubator supplemented with 5% CO_2_. The ST cells were maintained for 72 h to produce a cytopathic effect (CPE) in Dulbecco’s modified Eagle’s medium (DMEM) (Gibco, Billings, MT, USA) supplemented with 2% heat-inactivated fetal bovine serum (FBS) (Gibco). When obvious CPE appeared, cells were collected. The collected viruses were plaque-purified in 2 mL of DMEM containing 1% (*w*/*v*) low-melting-point agarose and 2% FBS, and their identity was validated by PCR as described previously [22].

### 2.4. Sequencing and Phylogenetic Analysis

The full-length gE gene was amplified by PCR from viral DNA extracted from the isolates as described previously [23]. The PCR product was purified using the V-ELUTE Gel Mini Purification Kit (Beijing Zoman Biotechnology Co., Ltd., Beijing, China), then ligated into the pMD^TM^ 18 T Vector Cloning Kit (Takara, Dalian, China), and finally transformed into Trelief^TM^ 5α Chemically Competent Cells (Tsingke, Beijing, China). The positive monoclonal clones were verified by PCR and sequenced by Wuhan AuGCT DNA-SYN Biotechnology Co., Ltd. (Wuhan, China), in triplicate. 

Twenty-eight PRV strains were retrieved from the NCBI database and served as the reference strains. The nucleotide sequences and the corresponding amino acid variations in the gE gene between PRV isolates sequenced in our study and the reference strains were analyzed using DNASTAR Lasergene.v7.1 (DNASTAR, Inc., Madison, WI, USA). The phylogenetic tree was constructed by the neighbor-joining method using the MEGA 7.0 software (www.megasoftware.net) with a bootstrap of 1000 replicates [24].

## 3. Results

### 3.1. Seroprevalence of PRV in Henan Province

In the present study, a total of 35,796 serum samples were collected in the Henan province, including 10,385 in 2019, 12,096 in 2020 and 13,315 in 2021. Meanwhile, 1090 involved farms were inoculated, including 324 in 2019, 385 in 2020 and 381 in 2021, from which sera were collected. Data obtained from the ELISA assay demonstrated that the positive rate at the serum sample level was 25.00% in 2019, 20.32% in 2020 and 16.69% in 2021 (Table 1), respectively, while at the farm level it was 50.31% in 2019, 50.65% in 2020 and 49.87% in 2021 (Table 2). Of the 1090 farms, 2278 serum samples were obtained from 98 small farms, 26,430 from 881 medium farms and 7088 from 111 large farms. The positive rate at small farms, medium farms and large farms was 25.72%, 22.15% and 11.78% at the serum sample level, and 69.39%, 48.81% and 45.05% at the farm level (Table 2). The seroprevalence of slaughterhouses, commercial pig farms and breeding farms was 25.83%, 23.80% and 8.14% (Table 1).

Regional variation, seasonal variation and other factors with a potential association with PRV were evaluated. From 2019 to 2021, the total seroprevalence was 14.01% in eastern Henan, 25.88% in western Henan, 17.68% in southern Henan, 20.39% in northern Henan and 23.13% in middle Henan (Figure 2a), revealing that PRV infection rates varied from region to region. From 2019 to 2021, the total seroprevalence rate was 21.70% in the first quarter (Q1), 24.16% in Q2, 21.38% in Q3 and 15.29% in Q4, implying PRV infection rates varied by quarter. In detail, the peak of PRV infection was 31.59% in Q3 of 2019 but 28.15% in Q1 of 2020 and 21.48% in Q2 of 2021, indicating that the peak time of PRV infection each year was inconsistent, while the lowest number of infections usually occurred in Q4 (Figure 2b). The performance in different distributions was more pronounced. During 2019 to 2021, the peak of PRV infection was 41.24% and 26.36% in Q1 in western Henan and northern Henan, compared to 30.36% in Q2 in middle Henan, and 19.52% and 26.87% in Q3 in eastern Henan and southern Henan (Figure 2c). Nevertheless, the lowest number of infections occurred in Q4, with 8.59% in the east, 22.32% in the west, 12.27% in the south, 12.04% in the north and 20.52% in the middle. Last but not least, the specific changes in the seroprevalence rates by quarter and by region are shown in Figure 2d, and the annual infection rate for each city is illustrated by Appendix A. 

### 3.2. The Results of PRV Isolation

The PRV gE gene was detected by PCR in 52 of the 389 (13.37%) clinical case samples. PCR-positive tissue samples were inoculated into ST cells, and distinct CPEs appeared after three to four blind passages on ST cells. PCR detection results confirmed that 18 PRV isolates were obtained, and these were named as HD-1, HD-2, PY-1, SC1, SC2, HN-CM, HN-CY, HN-GM, HN-HY, HN-LL, HN-LH, HN-LY, HN-YH, HN-HX, HN-MY, HN-WZ, HN-XT and HN-YY.

### 3.3. Genetic Analysis Results Based on gE

The full-length gE genes of 18 PRV isolates were cloned and sequenced. These sequences were submitted to GenBank under the accession numbers listed in Appendix A. Of the 18 full gE sequences examined, 12 isolates were 1737 nucleotides (nt) in length, encoding a protein of 578 amino acids (aas), and the other 6 isolates (HN-YH, PY-1, HD1, HD2, SC1 and SC2) were 1740 nt-long (579 aas). The nt sequencing analysis showed that the 18 isolates shared 98.6–99.9% nucleotide identity with each other, and these isolates shared 97.1% to 100.0% nucleotide sequence identity with 28 PRV reference strains (Appendix A). The phylogenetic analysis indicated that PRV strains could be divided into two genotypes (Figure 3), and these observations were corroborated by gE nt sequence identities of 18 PRV isolates in this study and 28 PRV reference strains. Genotype I included 10 PRV reference strains from Europe and America, and 18 PRV isolates in this study displayed 97.1% to 98.2% nucleotide sequence identity with the 10 PRV reference strains. All 18 PRV isolates in this study were clustered in genotype II, together with 12 Chinese PRV strains (after 2012), 3 early Chinese PRV strains (before 2012) and 3 Asian PRV strains, including the South Korean strain Yangsan, Japanese strain RC1 and Malaysian strain P-PrV, with nt sequence identities ranging from 98.7% to 99.9% between 18 PRV isolates in this study and the 18 reference strains. Phylogenetically, 6 of 18 PRV isolates (HN-YH, PY-1, HD1, HD2, SC1 and SC2) in this study and 10 Chinese variant PRV reference strains (after 2012) were distributed within the variant PRV cluster in genotype II and had 99.6–100.0% nucleotide sequence identity. The remaining 12 isolates, which represent the current PRV classical strains from a high-positive city at different times in each region in the Henan province, were located in the classical cluster with 2 early Chinese classical PRV reference strains, Ea and SC (before 2012), and 2 Chinese PRV reference strains, HuN-YY and HuB17 (after 2012), with nt sequence identity ranging from 99.1% to 99.9%. These results reveal that among the 18 isolates, HN-YH, PY-1, HD1, HD2, SC1 and SC2 belonged to PRV variants of genotype II, and the other 12 isolates were grouped as the classical PRV strains. In addition, combined with amino acid sequence analysis, unique aa variations in gE protein were used as molecular markers for differentiating gE clade divisions in genetic evolution analysis (Figure 3).

The aa sequences of gE protein in the 46 PRV strains were aligned, and all aa insertions and aa substitutions are depicted in Figure 4. Compared with genotype Ⅰ strains, 36 genotype Ⅱ strains had an aa insertion at position 48 (34D/ 2no insertion) and 18 aa substitutions at positions 59 (36D/Y→36N), 63 (36N→35D/1N), 106 (36V→36L), 122 (36A→35S/1A), 149 (36R→36M), 179 (36T→36S), 181 (36R/Q→36L), 215 (36L→36A), 216 (36A→36D), 472 (36G→36R), 474 (36R→36H), 504 (36A→35I/1V), 509 (36S→36A), 522 (36V→35A/1T), 526 (36A→36P), 573 (36S→35N/1S), 577 (36N→33M /2N/1H) and 578 (36A→33S/2A/1V).

## 4. Discussion

Due to dense swine populations, notifiable PR remains one of the most important diseases in regions of South America and some regions in Europe and Asia, especially in China, which is the largest producer of pork products in the world [3]. PR is recognized to have caused devastating damage to the Chinese swine industry and was listed in the “Middle-Long-term Animal Disease Prevention and Control Program in China (2012–2020)” [25], which aims to eradicate PR in pig farms in China by the end of 2020 (The State Council of the People’s Republic of China 2012). As a result, the prevalence of virulent PRV has declined significantly in China since 2012, but the eradication target has not yet been reached. As almost all pig farms in China are still using the gE-deleted PRV vaccine (Bartha K61 strain), detection of gE-specific antibodies by ELISA remains a rapid and effective method for differentiating between infected and vaccinated animals (DIVA) [26]. In this study, the overall positive rate for PRV gE antibodies decreased from 25.00% in 2019 to 16.69% in 2021, being significantly lower than 94.2% (49/52) in 2011 [18] and 30.14% (1419/4708) in 2018 [22]. It can be seen that the PR epidemic situation in the Henan province has exhibited an obvious downward trend, which is in agreement with the tendency of eradication. Recent epidemiological studies showed that the seroprevalence of PRV decreased from 38.20% in 2018 to 18.12% in 2020 in the Shandong province [27] from 62.40% in 2013 to 51.59% in 2018 in Tianjin [28] and from 20.9% in 2013 to 11.6% in 2018 in the Heilongjiang province [29], but increased from 19.91% in 2016 to 25.46% in 2020 in the Hunan province [30]. These data reveal that the positive rate of PRV infection in pigs differed in different geographical regions of China, which may have been caused by different situations of co-infection, vaccine protection, feeding management, introduction and quarantine, etc. The positive rate in farms in the Henan province was 50.31% (163/324) in 2019, 50.65% (195/385) in 2020 and 49.87% (190/381) in 2021 (Table 2). Associated epidemiological studies showed that PRV gE-antibody-positive rates of farms were 58.2% (124/213) in 23 regions of China [31], 49.76% (516/1037) from 2010 to 2018 in Tianjin [28] and 43.19% (349/808) in the Hunan province during 2016–2020 [30]. A possible reason for why the Henan province was more affected by the variant PR outbreaks is because some sows that were negative but could have transmitted PRV were not culled. Compared with other provinces, it took longer for the Henan province to complete the evolution of PRV. All data demonstrate that PRV is still circulating in Chinese pig herds.

In this study, the seroprevalence in small farms, medium farms and large farms was 25.72%, 22.15% and 11.78% at the serum sample level. This suggests the larger-scale pig farms may be more likely to benefit from stricter biosecurity measures, such as compulsory pig vaccination campaigns and sufficient regulations to reduce PR incidence and prevalence. In addition, the seroprevalence of serum samples from pig farms of different sizes (Figure 2e) and pig farms of different categories, including slaughterhouses, commercial pig farms and breeding farms (Figure 2f), declined year by year, implying that PRV was effectively controlled. It may be that the enhanced immune quality of PRV and the development of PR eradication have contributed to the effective control of pseudorabies. The immune quality of PRV, including the vaccine quality and vaccination density, has been improved, thereby increasing the resistance of pigs to the disease, which would be an effective factor for controlling the epidemic of PRV [26,32]. Regional prevention strategies should be adopted [32]. At the end of 2020, the statistics released by the Henan Survey Organization National Bureau of Statistics Information Network showed that the pig population exhibited a clear tendency toward south > west > central > north > east. Southern and western herds were larger but had greatly lower PRV-gE positivity, especially in Nanyang, at 8.06% (266/3301), in the south and Zhoukou, at 6.95% (302/4343), in the west, suggesting that swine farms in these regions may have developed or may be developing eradication of pseudorabies.

PRV seroprevalence was reported to be higher in autumn than in other seasons [22,30]. In this study, the highest seroprevalence rate was 24.16% (2095/8670) in Q2. This difference may be due to differences in sample size, time span or regional scope, resulting in insufficient data to determine the underlying causes of sudden fluctuations in a given location compared to a country. It was usually be speculated to be latent infection of the virus [8], co-infection with other pathogens [33] and cross-species transmission, etc. Furthermore, the seroprevalence rate in eastern Henan rose sharply from 0.00% (0/330) in Q1 to 49.17% (118/240) in Q2 in 2021, compared with the increase from 0.18% (1/570) in Q1 to 16.13% (121/750) in Q2 in 2019, which may be associated with the rate of 81.82% (90/110) seen in Q1 in 2021 in southern Henan. This suggests that the seasonal or seasonal infection may be influenced by the input and output of pigs and is easily misjudged as a temperature factor or others. Benefiting from sufficiently detailed data in the present study, the three sets of data in Table 1 may objectively explain this issue by the spread of the virus with the cross-regional transportation of live pigs. Additionally, seasonal and regional variations in PRV infection rates were evident which are in agreement with Sun’s study [16].

Nonetheless, no province has taken the lead in completing the eradication of PR. Regardless of near full vaccination rates, the virus, similar to other herpesviruses, can establish latent infection in the host’s peripheral nervous system via PRV in the field and reactivate after natural stimuli or stress factors, which is recognized as the most critical source of infection [15,34]. The currently available vaccines only provide partial protection against virulent PRV infection. Recombination of PRV strains in Suidae may result in changes in antigenicity, virulence and thus immune failure, which could be the source of continuing epidemics in China [35]. PRV may have spread from domestic pigs to dogs, and then to wild boars, in which the virus established itself and continues to circulate [36,37], and PRV may also cause bovine death through interspecies infection [38]. In particular, gE is a virulence factor of PRV infection in pigs and determines tropism for the central nervous system [39], which is frequently used to investigate the epidemiology and evolution of the virus [19,40]. Therefore, a serosurvey to detect specific antibodies against PRV and the etiological as well as genetic characteristics of PRV isolates would contribute to the rational use of vaccines and other novel viral inhibitors.

The phylogenetic analysis indicated that 10 PRV strains from Europe and America were identified as genotype I, whereas 18 PRV isolates in this study together with 12 Chinese PRV strains (after 2012), 3 early Chinese PRV strains (before 2012) and 3 Asian PRV strains, including South Korean strain Yangsan, Japanese strain RC1 and Malaysian strain P-PrV, were clustered in genotype II, which is in accordance with a previous report that most PRV isolates from China and other countries in Asia were classified into genotype II, whereas PRV strains from Europe and America were assigned to genotype I [41]. Interestingly, all 12 PRV isolates in this study and 1 PRV isolate reported in the Hunan province [30] belonged to classical PRV strains, meaning that the currently popular PRV strains could possibly recover to the rates of the classical strains, which requires further investigation. As previously shown, the nt insertions at 138–140 resulted in an aa insertion at position 48 (D) [16,20,22]. In this study, the aa insertion at position 48 (34D/ 2no insertion), which occurred in all genotype Ⅱ strains except the South Korean strain P-PrV from a pig and the Chinese strain Fa from a cow might be a characterization of the pseudorabies gE gene circulating in Asian pig herds after the first global outbreak. This finding confirms that PRV strains circulating in China evolved independently of strains isolated in Europe and America [42]. Furthermore, a cluster of 16 classical PRV strains has unique aa substitutions at positions 404 (16A→16P) and 520 (16P→15S/1P) compared with other clades. Surprisingly, the classical strains from the Henan province have a unique aa substitution at position 556 (12D→11G/1D) compared with other strains. Whether these mutations affect the virulence of PRV isolates currently circulating in China warrants further investigation.

Both an aa insertion at position 493 (15D/1G) and two aa mutations at positions 54 (16G→16D) and 449 (16V→15I/1V) were shared among the 16 PRV variants, which differ from other strains. A total of 10 out of 16 PRV variants (including 3 isolates in this study and 7 reference strains) had one substitution at position 512 (G→S). The nt insertions at positions 1472–1474 resulted in one aa insertion at position 493 (15D/1G), as also reported in previous studies [16,20,22], which represents a unique feature of variant PRV strains appearing after the pseudorabies outbreak in China in 2011. This supports the view that PRV strains in China may have evolved independently, leading to the emergence of variant strains [42]. Another feature found in the cluster of PRV variants was that nt substitutions at positions 161, 228 and 1345 resulted in aa substitutions at positions 54 (16G→16D), 449 (16V→15I/1V) and 512 (16G→10S/6G). Notably, the mutation at position 512 occurred in some of the variant strains, which indicated that PRV may still be mutating slowly. Due to the epitopes of gE protein localized at the aa positions 52–238 [43], the aa at position 54 (G→D) may be responsible for immune evasion. Previous studies showed novel PRV variants exhibited enhanced pathogenicity [35], and the variant strains HN1201, TJ, JS-2012 and hSD-1/2019 were more virulent and neurotropic to mice or pigs than the classical strains [13,17,44,45]. Whether the virulence of these isolates acquired in this study is enhanced remains to be further studied, involving proliferation characteristics in different types of macrophages, immune responses, pathogenicity to mice and pigs, etc.

In summary, the serological results demonstrate that that PR was still endemic at high levels in intensive pig herds in the Henan province, China, but displayed an obvious decreasing trend. Phylogenetic analysis based on the complete gE sequence found that both classical strains and variant strains existed in pig herds. Our finding of PR transmission through pigs transported across regions points to an inadequacy of PR prevention, and ongoing monitoring of PRV should be implemented for prevention and control.

## Figures and Tables

**Figure 1 viruses-14-01685-f001:**
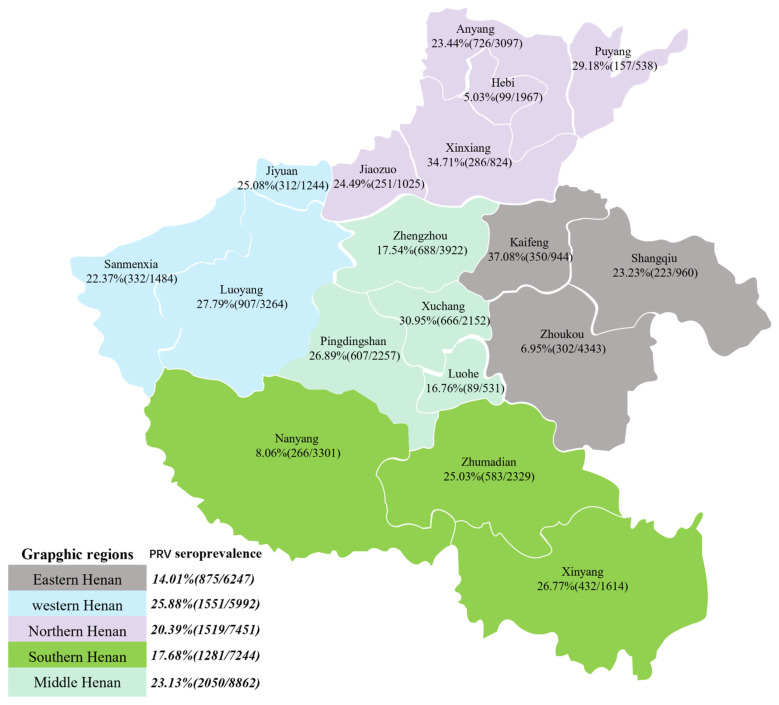
Seroprevalence of pseudorabies virus (PRV) in different geographical regions in the Henan province of China during 2019–2021.

**Figure 2 viruses-14-01685-f002:**
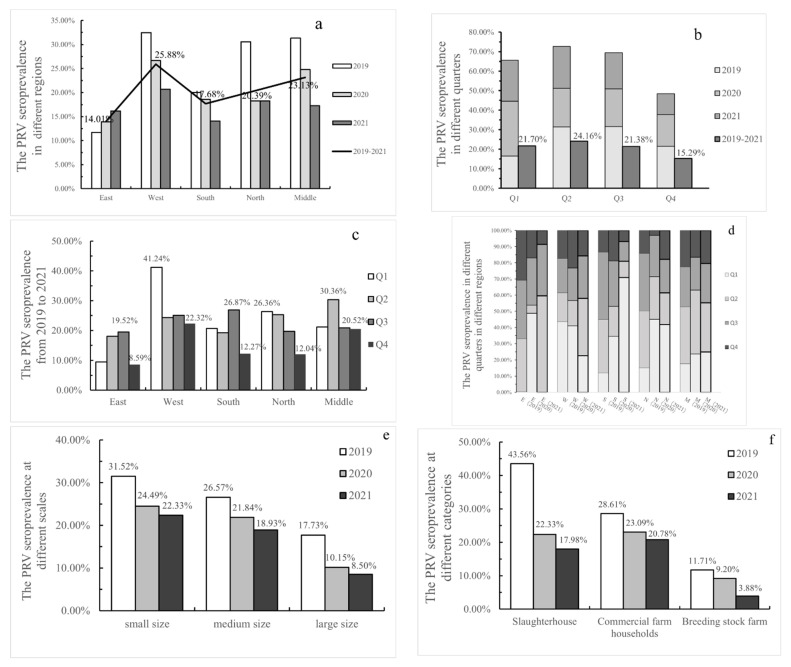
Seroprevalence rate of PRV-gE in the Henan province, China, 2019–2021, from different perspectives. (**a**) Seroprevalence rate of PRV-gE in different regions. (**b**) Seroprevalence rate of PRV-gE in different quarters. (**c**,**d**) Seroprevalence rate of PRV-gE in different quarters in different regions. (**e**) The PRV-gE seropositive rate of farms of different scales. (**f**) The PRV-gE seropositive rate of farms of different categories.

**Figure 3 viruses-14-01685-f003:**
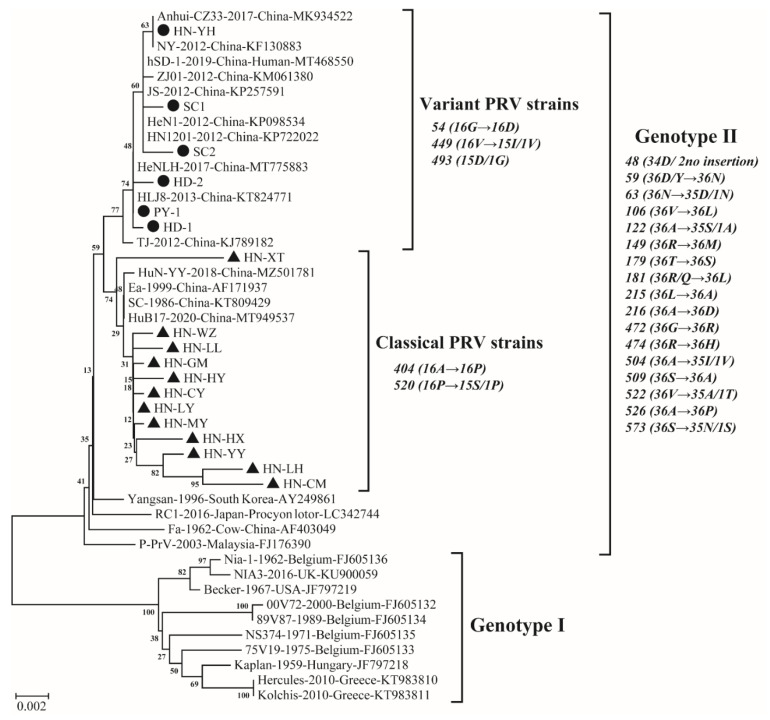
Phylogenetic tree based on the nucleotide sequences of gE gene of 18 PRV isolates determined in this study and 28 reference strains. Neighbor-joining trees were constructed with p-distance model and bootstrapping at 1000 replicates. Black solid circles (●) and black solid triangles (▲) represent variant strains and classical strains in this study, respectively.

**Figure 4 viruses-14-01685-f004:**
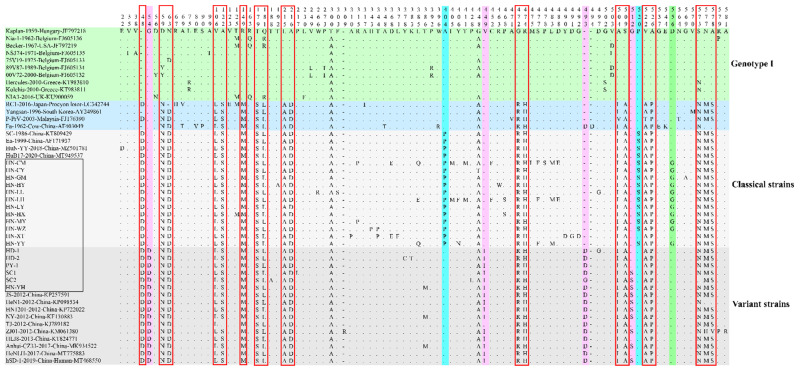
All amino acid mutation sites of gE protein of 18 PRV isolates sequenced in this study and 28 reference strains. All the strains were clustered into two genotypes, and genotype I only contained European–American PRV strains (mint green), while genotype II included Chinese variant strains (cloud gray), Chinese classical strains (pearl river gray) and Asian strains (Maya blue). The PRV isolates acquired in this study are shown in the hollow black rectangle. Compared with genotype I strains, unique amino acids substitutions of genotype II strains are shown in the hollow red rectangles. Amino acid mutations in variant strains are highlighted in lavender violet. The arctic blue areas represent unique amino acids substitutions in classical PRV strains. The screamin’ green area represents unique amino acid substitutions in classical PRV strains from the Henan province.

**Table 1 viruses-14-01685-t001:** The detection results for pseudorabies virus in different regions and different pig farms in the Henan province of China during 2019–2021.

Time	The Regions of Henan Province, China					The Categories of Pig Farms			Total
	Eastern Henan	Western Henan	Southern Henan	Northern Henan	Middle Henan	Slaughterhouse	Commercial Farm Households	Breeding Stock Farm	
January–March in 2019 (Q1)	0.18% (1/570)	61.92% (161/260)	11.40% (107/939)	18.78% (74/394)	22.13% (52/235)	62.50% (50/80)	27.50% (327/1189)	1.59% (18/1129)	16.47% (395/2398)
April–June in 2019 (Q2)	16.13% (121/750)	25.45% (112/440)	31.22% (153/490)	44.03% (295/670)	44.25% (200/452)	65.00% (195/300)	29.90% (572/1913)	19.35% (114/589)	31.44% (881/2802)
July–September in 2019(Q3)	17.79% (71/399)	30.11% (137/455)	39.21% (149/380)	44.13% (124/281)	31.12% (164/527)	46.86% (112/239)	27.96% (402/1438)	35.89% (131/365)	31.59% (645/2042)
October–December in 2019 (Q4)	15.22% (35/230)	24.67% (111/450)	12.69% (82/646)	17.76% (114/642)	28.34% (333/1175)	19.22% (69/359)	28.48% (508/1784)	9.80% (98/1000)	21.48% (675/3143)
Total for 2019	11.70% (228/1949)	32.46% (521/1605)	20.00% (491/2455)	30.55% (607/1987)	31.35% (749/2389)	43.56% (426/978)	28.61% (1809/6324)	11.71% (361/3083)	25.00% (2596/10,385)
January–March in 2020 (Q1)	24.42% (138/565)	52.94% (63/119)	26.96% (86/319)	29.97% (205/684)	24.11% (102/423)	33.47% (159/475)	29.45% (415/1409)	19.06% (61/320)	28.15% (594/2110)
April–June in 2020 (Q2)	2.56% (10/390)	20.05% (149/743)	14.53% (100/688)	17.59% (57/324)	40.00% (212/530)	14.76% (31/210)	25.76% (448/1739)	6.75% (49/726)	19.74% (528/2675)
July–September in 2020(Q3)	14.63% (109/745)	26.06% (135/518)	22.00% (187/850)	17.01% (272/1599)	20.64% (219/1061)	18.97% (184/970)	22.87% (705/3082)	4.58% (33/721)	19.32% (922/4773)
October–December in 2020 (Q4)	8.46% (33/390)	29.82% (201/674)	14.65% (103/703)	1.98% (7/353)	16.75% (70/418)	17.78% (16/90)	17.27% (328/1899)	12.75% (70/549)	16.31% (414/2538)
Total for 2020	13.88% (290/2090)	26.68% (548/2054)	18.59% (476/2560)	18.28% (541/2960)	24.79% (603/2432)	22.33% (383/1715)	23.09% (1862/8065)	9.20% (213/2316)	20.32% (2458/12,096)
January–March in 2021 (Q1)	0.00% (0/330)	18.67% (56/300)	81.82% (90/110)	27.89% (94/337)	17.20% (65/378)	18.64% (11/59)	21.88% (258/1179)	16.59% (36/217)	20.96% (305/1455)
April–June in 2021 (Q2)	49.17% (118/240)	29.12% (166/570)	11.71% (41/350)	13.16% (109/828)	20.91% (252/1205)	14.11% (59/418)	25.28% (565/2235)	11.48% (62/540)	21.48% (686/3193)
July–September in 2021(Q3)	26.29% (168/639)	21.65% (176/813)	14.04% (99/705)	13.84% (62/448)	16.82% (215/1278)	18.67% (84/450)	23.90% (625/2615)	1.34% (11/818)	18.54% (720/3883)
October–December in 2021 (Q4)	7.11% (71/999)	12.92% (84/650)	7.89% (84/1064)	11.90% (106/891)	14.07% (166/1180)	20.42% (106/519)	13.95% (398/2854)	0.61% (7/1141)	10.68% (511/4784)
Total for 2021	16.17% (357/2208)	20.66% (482/2333)	14.09% (314/2229)	18.28% (371/2504)	17.27% (698/4041)	17.98% (260/1446)	20.78% (1846/8883)	3.88% (116/2986)	16.69% (2222/13,315)
Total for 2019–2021	14.01% (875/6247)	25.88% (1551/5992)	17.68% (1281/7244)	20.39% (1519/7451)	23.13% (2050/8862)	25.83% (1069/4139)	23.80% (5517/23,177)	8.14% (690/8480)	20.33% (7276/35,796)

Note: Q represents quarter.

**Table 2 viruses-14-01685-t002:** The prevalence of pseudorabies virus in different-scale pig herds in the Henan province during 2019–2021.

Year	Prevalence on Pig Farms of Different Sizes				Prevalence of Serum Samples on Pig Farms of Different Sizes			
	Small (15∼29)	Medium (30)	Large (31∼200)	Total	Small	Medium	Large	Total
2019	73.33% (22/30)	48.81% (123/252)	42.86% (18/42)	50.31% (163/324)	31.52% (197/625)	26.57% (2009/7560)	17.73% (390/2200)	25.00% (2596/10,385)
2020	69.23% (27/39)	49.20% (154/313)	42.42% (14/33)	50.65% (195/385)	24.49% (226/923)	21.84% (2051/9390)	10.15% (181/1783)	20.32% (2458/12,096)
2021	65.52% (19/29)	48.42% (153/316)	50.00% (18/36)	49.87% (190/381)	22.33% (163/730)	18.93% (1795/9480)	8.50% (264/3105)	16.69% (2222/13,315)
Total	69.39% (68/98)	48.81% (430/881)	45.05% (50/111)	50.28% (548/1090)	25.72% (586/2278)	22.15% (5855/26,430)	11.78% (835/7088)	20.33% (7276/35,796)

## Data Availability

The accession numbers presented in this study can be found in online repositories (https://www.ncbi.nlm.nih.gov/) accessed on 24 May 2022.

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
