# Peer review of "Serological Investigation and Genetic Characteristics of Pseudorabies Virus between 2019 and 2021 in Henan Province of China"

_viruses, 2022, doi:10.3390/v14081685_

Round 1

Reviewer 1 Report

The research by Chen and his colleagues has important guiding significance for the epidemic research of PRV in Henan Province. They examined PRV gE-specific antibodies in pigs from 2019 to 2021 on pig farms to determine the prevalence of PRV in Henan Province under Bartha-K61 vaccination. The results revealed that the overall positive rate for PRV gE antibody was 20.33% (7276/35796), which decreased from 25.00% (2596/10385) in 2019 to 16.69% (2222/13315) in 2021, demonstrating that PR still existed widely in pig herds in Henan Province but displayed a decreasing trend. In addition, the gE gene was amplified from the positive samples by strain isolation and analyzed by evolution. The phylogenetic analysis based on the gE complete sequences found that there were both classical strains and variant strains in pig herds. The deduced amino acid sequence analysis of gE gene showed that there were typical amino acids in the classical strains, the variant strains, and Genotype â…¡ strains, respectively. Although there is preliminary evidence that the prevalence of PRV shows a declining trend , there are several issues with this manuscript, as I highlight below:

1. The authors collected pig serum from pig farms in Henan Province since 2019 to detect PRV gE-specific antibody. I wondered how the authors to ensure the validity of the data? In 2019, the Sera from vaccinated pigs were started to collect, even stored at -80°C, could not be guaranteed to have no effect on the samples, not to mention the fact that the authors were testing serum samples for gE-specific antibodies in 2021. Whether or not this would affect the validity of the data, the author needs to indicate the time of sampling and the specific time of detection, and list the quality control standards applied in this study to ensure the authenticity and validity of the data.

2. In Discussion section, authors listed a series of data about PRV gE-specific antibodies in different provinces of China. There are some differences between the data. How can the authors explain this regional difference, and the positive rate of Henan Province is obviously higher than that of other provinces except Tianjin (The farm positivity rate in Henan Province was 50.31% (163/324) in 2019, 50.65% (195/385) in 2020, and 49.87% (190/381) in 2021 (Table 2))? What are the possible reasons?

3. Should take attention to the format and blank space. For example, Line 146-147, 18 full gE sequences examined, 12 strains were...and other six strains (HN-YH...); Line 169, the other twelve isolates...Figures should be presented in a consistent manner.

Reviewer 2 Report

This manuscript reports serological study of PRV in Henan province of China. The presentation of data is extremely poor. The data should be reported in table or figure form instead of piling hundreds of number in text format, making it unreadble. I would ask the authors to revise to improve data representation.

Round 2

Reviewer 1 Report

Accept in present form